# Association between Early Mobilization in the ICU and Psychiatric Symptoms after Surviving a Critical Illness: A Multi-Center Prospective Cohort Study

**DOI:** 10.3390/jcm11092587

**Published:** 2022-05-05

**Authors:** Shinichi Watanabe, Keibun Liu, Kensuke Nakamura, Ryo Kozu, Tatsuya Horibe, Kenzo Ishii, Daisetsu Yasumura, You Takahashi, Tomoya Nanba, Yasunari Morita, Takahiro Kanaya, Shuichi Suzuki, Alan Kawarai Lefor, Hajime Katsukawa, Toru Kotani

**Affiliations:** 1Department of Rehabilitation, National Hospital Organization, Nagoya Medical Center, 4-1-1 Sannomaru, Naka-ku, Nagoya 460-0001, Japan; billabonghonor@yahoo.co.jp (S.W.); billabonghono@yahoo.co.jp (T.K.); 2Department of Physical Therapy, Faculty of Rehabilitation, Gifu University of Health Science, 2-92 Higashiuzura, Gifu 500-8281, Japan; 3Critical Care Research Group, The Prince Charles Hospital, 627 Rode Rd, Chermside, QLD 4032, Australia; 4Department of Emergency and Critical Care Medicine, Hitachi General Hospital, 2-1-1 Jounann, Hitachi 317-0077, Japan; mamashockpapashock@yahoo.co.jp; 5Department of Cardiopulmonary Rehabilitation Science, Nagasaki University Graduate School of Biomedical Sciences, 1-14 Bunkyou-cho, Nagasaki 852-8521, Japan; ryokozu@nagasaki-u.ac.jp; 6Department of Rehabilitation, Tokyo Women’s Medical University, 8-1 Kawata-cho, Shinjuku-ku, Tokyo 162-8666, Japan; king.of.rehabilly@gmail.com; 7Department of Anesthesiology, Intensive Care Unit, Fukuyama City Hospital, 3-8-5 Zao-cho, Fukuyama 721-8511, Japan; keishii1101@gmail.com; 8Department of Rehabilitation, Naha City Hospital, 2-31-1 Furujima, Naha 902-8511, Japan; yasumuradai@yahoo.ne.jp; 9Department of Healthcare Administration, The University of Kyushu, 3-1-1, Maidashi, Higashi-ku, Fukuoka 812-8582, Japan; 10Department of Rehabilitation, Yuuai Medical Center, 50-5 Yone, Tomigusuku 901-0224, Japan; yo.takahashi7448@gmail.com; 11Department of Rehabilitation, Yao Tokushukai General Hospital, 1-17 Wakakusachou, Yao-shi, Osaka 581-0011, Japan; nanbatomoya@yahoo.co.jp; 12Department of Critical Care Medicine, National Hospital Organization, Nagoya Medical Center, 4-1-1 Sannomaru, Naka-ku, Nagoya 460-0001, Japan; moltlyme2@yahoo.co.jp (Y.M.); oltlyme2@yahoo.co.jp (S.S.); 13Department of Surgery, Jichi Medical University, 3311-1 Yakushiji, Shimostuke-shi 329-0498, Japan; alefor@jichi.ac.jp; 14Japanese Society for Early Mobilization, 1-2-12 Kudankita, Tiyoda-ku, Tokyo 102-0073, Japan; winegood21@gmail.com; 15Department of Intensive Care Medicine, School of Medicine, Showa University, 1-5-8 Hatanodai, Shinagawa-ku, Tokyo 142-8666, Japan; trkotani@med.showa-u.ac.jp

**Keywords:** anxiety, early mobilization, depression, ICU care, mental health, post-traumatic stress disorder

## Abstract

This is a prospective multicenter cohort study aiming to investigate the association between early mobilization (EM), defined as a rehabilitation level of sitting at the edge of the bed or higher within 72 h of ICU admission, and psychiatric outcome. Consecutive patients, admitted to the ICU for more than 48 h, were enrolled. The primary outcome was the incidence of psychiatric symptoms at 3 months after hospital discharge defined as the presence of any of three symptoms: depression, anxiety, or post-traumatic stress disorder (PTSD). Risk ratio (RR) and multiple logistic regression analysis were used. As a sensitivity analysis, two methods for inverse probability of treatment weighting statistics were performed. Of the 192 discharged patients, 99 (52%) were assessed. The patients who achieved EM had a lower incidence of psychiatric symptoms compared to those who did not (25% vs. 51%, *p*-value 0.008, odds ratio (OR) 0.27, adjusted *p* = 0.032). The RR for psychiatric symptoms in the EM group was 0.49 [95% Confidence Interval, 0.29–0.83]. Sensitivity analysis accounting for the influence of death, loss to follow-up (OR 0.28, adjusted *p* = 0.008), or potential confounders (OR 0.49, adjusted *p* = 0.046) consistently showed a lower incidence of psychiatric symptoms in the EM group. EM was consistently associated with fewer psychiatric symptoms.

## 1. Introduction

Critical illness may result in severe psychiatric disorders, such as depression, anxiety, and post-traumatic stress disorder (PTSD) [1,2], which adversely affect the quality of life and prevent intensive care unit (ICU) survivors from returning to their original lives [3]. Psychiatric symptoms occur in 10–70% of ICU survivors [1,2,4,5] and could last for several months or years after hospital discharge [6,7]. Although research interest is growing as shown by the increasing body of literature, there is no strategy with definitive effects to prevent the development of these symptoms.

Immobility induced by physical restriction is associated with the development of psychiatric disorders [8,9], while the potential benefits of regular physical exercise in non-ICU settings to decrease psychiatric symptoms have been described [10,11]. Active physical rehabilitation during the ICU stay, especially when initiated within the first 72 h of ICU stay, is recommended to prevent physical disabilities and improve clinical outcomes of ICU patients [12]. However, the effects of physical rehabilitation early in the ICU stay on psychiatric symptoms are unknown in the existing literature [13].

Therefore, we conducted a multicenter prospective cohort study to investigate the incidence rate of psychiatric symptoms at 3 months after hospital discharge and the association between active physical rehabilitation within 72 h of ICU admission and psychiatric symptoms. We focused on 3 months since a similar trend and incidence rate of psychiatric symptoms at 3 months and 1 year after hospital discharge have been observed [14,15].

## 2. Materials and Methods

### 2.1. Study Design and Patient Selection

This multicenter prospective cohort study was approved by the Ethics Committee of Nagoya Medical Center (2018093) and eight other participating hospitals (Hitachi General Hospital, Nagasaki University Hospital, Fukuyama City Hospital, Naha City Hospital, Yuuai Medical Center, Tokushukai General Hospital, Showa University Hospital, Tokyo Women’s Medical University Hospital) and registered in UMIN (ID: 000036503). We followed the STROBE guidelines [16], and all methods in this study were carried out in accordance with relevant guidelines and regulations. Informed consent was obtained from all patients.

Consecutive patients, up to 25 patients in each participating hospital, who stayed in the ICU for more than 48 h between June and December in 2019, were eligible for enrollment. Patients less than 18 years of age, unable to walk independently before admission, with neurological complications, lacking communication skills due to pre-existing mental diseases, or in a terminal state were excluded (Appendix A). Our study excludes patients with a history of psychiatric disorders (depression, anxiety, PTSD). Patients who died or did not complete the assessment at 3 months follow-up after hospital discharge, were also excluded. In the present study, mobilization was defined as physical rehabilitation at the level of sitting on the edge of the bed or higher [17], and patients were divided into two groups. The patients were allocated into the early mobilization (EM) group when they achieved rehabilitation at the level of mobilization within 72 h of ICU admission [17,18,19]. Patients were allocated into the non-EM group when they did not receive mobilization during the ICU stay or achieved mobilization more than 72 h after ICU admission. All patients equally aimed to receive mobilization daily under a protocol tailored to the circumstances of participating hospitals, though mobilization could not be achieved when the patient did not meet the criteria (Appendix A). Patients were not randomized in this study, and whether patients could undergo mobilization depended on the EM protocol used in each participating hospital.

### 2.2. EM Protocol

The early goal-directed protocol for rehabilitation [19,20,21] was developed more than 6 months before this study was initiated, and the details of the contents were arranged based on the situation of participating hospital. This protocol is used in multiple centers in routine practice and safety validation has already been reported [22]. The EM protocol details are shown in Appendix A.

The protocol provides patients with five rehabilitation levels (level 1, passive range of motion and respiration physical therapy; level 2, active range of motion; level 3, sitting exercise; level 4, standing exercise; and level 5, walking exercise) based on their medical condition. At each participating site, ICU physicians or physiotherapists made the final decision of the rehabilitation level based on the patient’s condition while referring to the protocol. All patients were supposed to receive at least one rehabilitation session per day for 20 min on weekdays.

Regarding other ICU care, all participating hospitals followed the 2018 Clinical Practice Guidelines [23] and the clinical practice guideline for the management of ARDS [24].

After transfer out of the ICU, all patients underwent rehabilitation, such as muscle strengthening, balancing, walking, and stair exercises, for more than 20 min on weekdays by physical or occupational therapists according to the rehabilitation policy in the general ward of each hospital without any specific protocol.

### 2.3. Follow-Up Protocol

Appendix A shows the details of the follow-up protocol. At 3 months after hospital discharge, a doctor, a nurse, or physiotherapist at each participating hospital made a phone call to the patient to confirm their survival. Once survival was confirmed, evaluation of the EuroQol-5 Dimensions-5 Levels (EQ-5D-5L) [25] was verbally performed during the same phone call. Then, questionnaires and response sheets for Hospital Anxiety and Depression Scale (HADS) and Impact of Events Scale-Revised (IES-R) were mailed to the patient’ home. The questionnaires were sent back to each participating hospital with the patient’s responses. Despite several phone calls, if the patient did not respond, they were excluded from analysis as lost to follow-up.

### 2.4. Data Collection

Baseline characteristics were collected at the time of ICU admission and during the ICU stay by co-investigators at each hospital, including age, gender, body mass index, Charlson comorbidity index [26], Barthel Index before hospitalization [27], ICU admission diagnosis, Acute Physiology and Chronic Health Evaluation II score, Sequential Organ Failure Assessment score, and use of mechanical ventilation, continuous vasopressors, continuous analgesia, continuous sedation, steroids, neuromuscular blocking agents, dialysis, time to first out of bed mobilization from the time of ICU admission, highest ICU Mobility Scale (IMS) during ICU stay, and the number of daily rehabilitation sessions on the ward. Barthel index before hospitalization was scored at the time of ICU admission based on the information from the family or the patients if they were conscious. The average sedation level, described according to the Richmond Agitation-Sedation Scale (RASS), from days 1 to 3 was calculated based on data in the electronic medical record. The IMS provides a sensitive 11-point ordinal scale, ranging from nothing (lying/passive exercises in bed, score of 0) to independent ambulation (score of 10) [28].

### 2.5. Study Outcomes

The primary outcome was the incidence of psychiatric symptoms at 3 months after hospital discharge, which was defined as the presence of at least one of three symptoms: depression, anxiety, or PTSD. Depression and anxiety were assessed using the HADS that contains 14 items, seven for anxiety assessment and seven for depression, with a score of 0–3 for each item. Within a maximum score of 21 for each subset for depression or anxiety assessment, the presence of depression or anxiety was defined as a score of 8 or more [29]. PTSD was evaluated using the IES-R, a 22-item self-reported measure with scores ranging between 0 and 88 points (score 0–4 per item). The presence of PTSD was defined as a score of 25 or more [30].

Secondary outcomes were the scores on HADS subsets for depression or anxiety and the IES-R score for PTSD at 3 months after hospital discharge, as well as the change in each score between hospital discharge and at 3 months.

Other variables included the EQ-5D-5L which is a standardized assessment for HRQoL [25] at 3 months follow-up and at hospital discharge, walking independence at discharge [21], duration of mechanical ventilation, length of ICU and hospital stays, Barthel Index at hospital discharge, incidence of delirium during ICU stay, incidence of ICU-acquired weakness (ICU-AW) at ICU discharge. Patients who could walk 45 m or more with or without braces were regarded as walking independent [19]. For the assessment of delirium, either Confusion Assessment Method for the Intensive Care Unit [31] or Intensive Care Delirium Screening Checklist [32] was used as delirium screening tool. ICU Acquired Weakness is defined that Medical Research Council-sum score evaluating by physical therapists is less than 48 at the time of ICU discharge [33].

### 2.6. Statistical Analysis

A sample size of 240 patients is needed with 80% power and a two-sided significance level of 0.05, under the assumption of a follow-up achievement rate of 80%, 60% of non-EM patients will develop psychiatric symptoms and EM patients will have a 20% reduction based on previous studies [14,15,34].

Data are presented as a median with interquartile range or as a number with percentage. The Mann–Whitney U test was used to analyze continuous variables and the χ^2^ test or Fisher’s exact test for nominal variables, as appropriate. Before using a non-parametric test, the distribution of each parameter was evaluated with the Shapiro–Wilk test.

In addition to the comparison of baseline characteristics between the two groups, given the influence of death and loss to follow-up, we also compared them between patients discharged from the ICU and those who completed follow-up at 3 months. The same analysis was conducted among patients discharged from ICUs and those who completed follow-up both in the EM and non-EM groups.

Multiple logistic regression analysis was performed to identify an association of the primary outcome with the following covariates: Age, male gender, Barthel index before hospitalization, ICU admission diagnosis, APACHE II score, and use of mechanical ventilation, continuous analgesia, continuous sedation, steroids, and neuromuscular blocking agents, and dialysis, which were considered as factors related to the primary outcome in previous reports (Appendix A) [35,36,37,38,39,40,41,42,43,44,45,46,47]. To analyze secondary and other outcomes, multiple linear and logistic regression analyses were performed for log-transformed continuous and categorical variables, respectively, using the same covariates as used in analyzing the primary outcome. Each relative risk ratio (RR) for the incidence of depression, anxiety, or PTSD in the EM group against the non-EM group was described. Pearson’s correlation coefficient was used to assess the correlation between the number of days from ICU admission to first mobilization and the HADS subset scores for depression or anxiety or the IES-R score for PTSD.

As a post hoc analysis, two methods for inverse probability of treatment weighting (IPTW) statistics were used. The first adjusted the influence of death and follow-up loss and the second adjusted as many confounding factors from baseline characteristics such as severity, baseline comorbidity, ICU admission diagnosis, consciousness level, which could affect the outcomes. The methodological details of the analysis are shown in Appendix A [48,49].

All analyses were performed using SPSS version 23.0 (IBM Corp., Armonk, NY, USA). Statistical tests were two-sided, and statistical significance was defined as *p*-values of <0.05.

## 3. Results

### 3.1. Baseline Characteristics

Of the 1014 patients screened, 203 were enrolled (Figure 1). Of these, 192 patients were discharged from the ICU and were eligible for follow-up assessment. Excluding patients who died or who missed the 3-month follow-up, a total of 99 patients, including 60 in the EM group and 39 in the non-EM group, completed the 3-month follow-up. There were differences among the measured baseline characteristics between the two groups, for the Acute Physiology and Chronic Health Evaluation II score (17 vs. 21, *p* = 0.026) and the use of mechanical ventilation (53% vs. 74%, *p* = 0.034), steroids, neuromuscular blockade (0% vs. 13%, *p* = 0.008), and dialysis (12% vs. 28%, *p* = 0.037) (Table 1). The average RASS score within the first 3 days of the ICU stay was not significantly lower in the EM group. Among the rehabilitation items, the EM group had a shorter time from ICU admission to first rehabilitation (1.7 days vs. 5.3 days, *p* < 0.001), and a higher IMS score during their ICU stay (8 vs. 4, *p* < 0.001).

Considering the bias due to death and loss to follow-up, no differences were observed in all baseline characteristics comparing patients discharged from the ICU and who completed the 3-month follow-up in each comparison (Appendix A).

### 3.2. Primary and Secondary Outcomes

The incidence of psychiatric symptoms at the 3-month follow-up point was significantly lower in the EM group than in the non-EM group [odds ratio (OR): 0.27, adjusted *p* = 0.032] even after adjustment for the baseline characteristics of the two groups, whereas the incidence at the time of hospital discharge was not different (OR: 1.03, adjusted *p* = 0.965) (Table 2).

At 3 months follow-up, the EM group showed a significantly lower incidence of PTSD (OR: 0.06, adjusted *p* = 0.026) and had a significantly lower HADS subset score for anxiety (adjusted *p* = 0.004) and IES-R (adjusted *p* = 0.009) compared with the non-EM group. The EM group demonstrated no significant differences in the incidence of depression, anxiety, and PTSD, the HADS subset scores for depression and anxiety, and the IES-R score at the time of hospital discharge. When comparing the assessment scores at hospital discharge and at 3 months follow-up, only changes in the HADS subset scores for anxiety in the EM group were significantly higher (adjusted *p* = 0.032).

The risk for developing psychiatric symptoms [RR: 0.49, confidence interval (CI): 0.29–0.83, *p* = 0.010], depression (RR: 0.52, CI: 0.27–0.99, *p* = 0.006), anxiety (RR: 0.27, CI: 0.10–0.71, *p* < 0.001), and PTSD (RR: 0.07, CI: 0.01–0.54, *p* < 0.001) at 3 months follow-up were lower in the EM group, while no significant difference in the risk ratios was observed at the time of hospital discharge (Table 3).

The time from ICU admission to first mobilization correlated weak linearly with the HADS subset scores for depression (r = 0.244, *p* = 0.011) and anxiety (r = 0.350, *p* < 0.001), and the IES-R score (r = 0.289, *p* = 0.003) at 3 months follow-up (Appendix A). Multiple linear regression analysis also demonstrated that EM was significantly associated with the HADS subset score for anxiety [β coefficient: 0.311, adjusted *p* = 0.004] and the IES-R score (β coefficient: 0.278, adjusted *p* = 0.009) at 3 months follow-up. Changes in the HADS subset scores for anxiety were significantly associated with achieving EM (β coefficient: −0.242, adjusted *p* = 0.032) (Appendix A).

### 3.3. Other Variables

There was no difference between the two groups in the EQ-5D-5L at 3 months follow-up and at hospital discharge. The EM group had a shorter length of ICU and hospital stays, a higher Barthel Index at hospital discharge, and a lower incidence of delirium and ICU-AW. (Table 4).

### 3.4. Post Hoc Sensitivity Analysis

The model created by IPTW considering the influence of death and follow-up loss showed that EM correlated significantly with a lower incidence of psychiatric symptoms (OR: 0.28, CI: 0.11–0.73, adjusted *p* = 0.008), lower incidence of anxiety and PTSD, and a lower HADS subset score for anxiety and IES-R score at 3 months follow-up (Appendix A). In the IPTW analysis using the propensity score with adjustment of 19 baseline variables, EM correlated significantly with a lower incidence of psychiatric symptoms at 3 months (OR: 0.49, CI: 0.14–0.93, adjusted *p* = 0.046) and a lower incidence of PTSD and a lower HADS subset score for anxiety and IES-R score (Appendix A).

## 4. Discussion

The incidence of psychiatric symptoms was significantly lower in the EM group and EM in the ICU was significantly associated with a lower incidence of psychiatric symptoms at 3 months follow-up after hospital discharge even after adjustment for significant differences in the baseline characteristics including severity and treatments received. The results were consistent even if the influence of patient death, loss to follow-up or potential confounding factors were included in the analysis. Furthermore, the interval from ICU admission to first mobilization correlated significantly with the score of HADS subsets for Depression and Anxiety and the score of IES-R for PTSD.

Mobilization, rehabilitation at the level of sitting on the edge of the bed or higher was significantly associated with improved psychiatric symptoms. A previous systematic review using mobilization in the ICU as an intervention did not show a significant improvement in psychiatric symptoms after ICU discharge [50]. Of note, most studies in that systematic review did not define the time point for mobilization, despite the potential benefit of mobilization within 72 h of ICU admission (Early Mobilization) which is described in current literature [13,51]. For instance, providing EM improves independent physical function at the time of discharge, shortens delirium duration, and increases ventilator-free days [52], whereas mobilization initiated more than 72 h after ICU admission has no beneficial effect [13,53].

The present study supports the idea that the effect of EM on outcomes related to psychiatric symptoms after ICU discharge as well as other clinical outcomes might be maximized if mobilization is initiated within 72 h of ICU admission. The significant correlation between time from ICU admission to achieve EM and scores for diagnosing psychiatric symptoms support this idea. The interval until initiation of mobilization might be an essential aspect of physical rehabilitation in the ICU to maximize the effect on patient outcomes.

The EM group had a lower incidence of psychiatric symptoms at 3 months follow-up, but there was no significant difference between the groups at discharge. It has been reported that there is a time lag for the onset of anxiety and depression [54,55]. Therefore, the time of onset of each psychiatric symptom, such as depression, anxiety, and PTSD, could be different [56]. The time for the effect of EM on outcomes after hospital discharge to appear could vary among the psychiatric symptoms. Comparing the difference at 3 months and at hospital discharge, anxiety was the most improved among the psychiatric symptoms assessed. As previous studies have shown, an exercise period of at least 8 weeks is required to reduce depression, whereas anxiety disorders are improved with exercise in a relatively short period [13,57]. Considering these, the follow-up period might be adjusted depending on outcomes and interventions of the study related to psychiatric symptoms after ICU admission.

Several studies suggest an interaction between physical and psychiatric factors and these outcomes could be linked [58,59]. Physical impairment could result in poor psychiatric function and vice versa. For example, ICU-AW could lead to the development of delirium which is a potential risk factor for the development of psychiatric disorders [60]. There is evidence to support the idea that EM could reduce not only physical impairment, such as ICU-AW, but also delirium similar to the results of the present study [61] and might result in a synergistic and beneficial effect on psychiatric function. It has been suggested that exercise or mobilization improves mental health [62] and self-efficacy [63], increase endorphin and monoamine levels [63], decrease cortisol levels [64], and increase brain-derived neurotrophic factor [65], resulting in better psychiatric outcomes as a potential mechanism for mobilization improving psychiatric symptoms. In addition, there were several differences in outcomes at the time of hospital discharge, such as physical function, that could be not only intervening factors but also mediators of psychiatric symptoms after a critical illness. Further studies are warranted to validate the relationship between physical and psychiatric function, and the background mechanisms to capture the interaction between interventions such as mobilization, physical impairment at hospital discharge, and psychiatric symptoms at follow-up.

The comparability between the two groups is a primary limitation of this study. The calculated sample size could not be achieved, there was a relatively large influence of death and loss to follow-up and the study was not randomized. These factors may limit generalizability to other ICUs. Second, there are confounding factors which could not be adjusted for, such as educational history and clinical frailty. However, multivariate analysis with important clinical factors, IPTW, and propensity score matching adjusted by potential confounders showed consistent results even though there are several differences in important baseline characteristics of the two groups, for example the use of mechanical ventilation, steroids, neuromuscular blocking agents and dialysis. Third, outcomes were limited to short-term follow-up. Fourth, whether the patient could receive rehabilitation at the level of sitting on the edge of the bed or higher depended on the rehabilitation policy used in each participating hospital. Therefore, whether EM could not be provided due to poor general condition or other factors was not identified. Fifth, the psychiatric symptoms of the primary outcome of this study are diagnosed by scoring and are not based on criteria by psychiatrists or psychiatric liaison teams. Finally, patients who died in the ICU were excluded from the analysis in this study because we targeted ICU survivors. This could result in selection bias and these findings should be interpreted with caution. To further validate these results and investigate causality, a multicenter randomized controlled trial with more patients is needed.

## 5. Conclusions

EM in the ICU is significantly associated with lower rates of psychiatric symptoms, including depression, anxiety, and PTSD, at 3 months follow-up after hospital discharge. The interval from ICU admission to mobilization might be an important parameter to maximize the beneficial effects on patient outcomes.

## Figures and Tables

**Figure 1 jcm-11-02587-f001:**
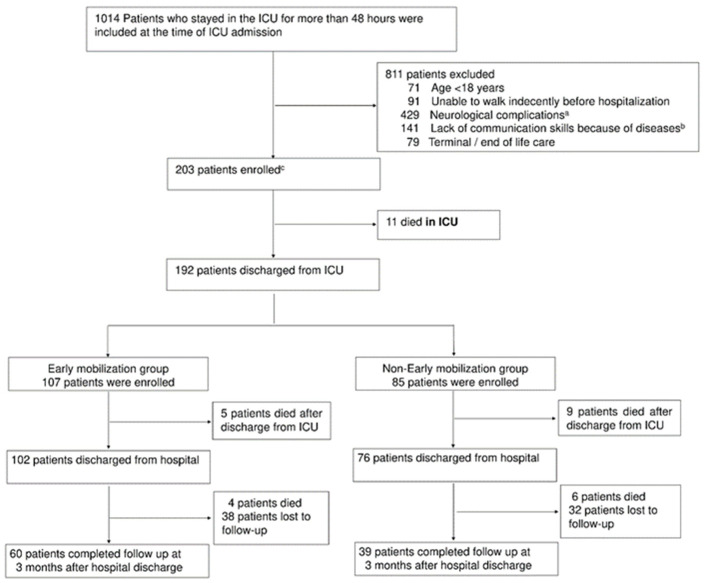
Study flow chart. ICU = intensive care unit. ^a^ Neurological complications include cerebral infarction, cerebral hemorrhage, acute subdual hematoma, acute epidural hematoma, traumatic subarachnoid hemorrhage, and encephalitis. ^b^ Diseases include depression, anxiety, schizophrenia, dementia, cerebral infarction, cerebral hemorrhage, and alcoholism. ^c^ Four out of nine hospitals could not enroll 25 patients and enrolled 14, 21, 22, and 21 patients.

**Table 1 jcm-11-02587-t001:** Baseline characteristics.

Variables	Early Mobilization Group (*n* = 60)	Non-Early Mobilization Group (*n* = 39)	*p* Value
Age (years), median (IQR)	70 (61–75)	73 (57–79)	0.416
Gender (male), *n* (%)	39 (65)	23 (59)	0.545
BMI (kg/m^2^), median (IQR)	23 (21–25)	24 (21–27)	0.439
Charlson Comorbidity Index, median (IQR)	2 (1–2)	1 (0–2)	0.591
Barthel index before hospitalization, median (IQR) ^a^	100 (100–100)	100 (100–100)	0.638
ICU admission diagnosis, *n* (%)			0.652
Acute respiratory failure (including pneumonia)	6 (10)	6 (15)	
Cardiovascular disease	30 (50)	18 (46)	
Gastric or colonic surgery	10 (17)	4 (10)	
Sepsis, non-pulmonary	9 (15)	5 (13)	
Other diagnoses	5 (8)	6 (15)	
APACHE II score, median (IQR)	17 (12–22)	21 (16–26)	0.026
SOFA at ICU admission, median (IQR)	7 (3–8)	7 (4–11)	0.121
The use of mechanical ventilation during ICU stay, *n* (%)	32 (53)	29 (74)	0.034
The use of continuous vasopressor during ICU stay, *n* (%)	34 (57)	25 (64)	0.461
The use of continuous analgesia during ICU stay, *n* (%)	37 (62)	26 (67)	0.613
The use of continuous sedation during ICU stay, *n* (%)	45 (75)	28 (71)	0.723
The use of steroid during ICU stay, *n* (%)	7 (12)	14 (35)	0.006
The use of neuromuscular blocking agent during ICU stay, *n* (%)	0 (0)	5 (13)	0.008
The use of dialysis during ICU stay, *n* (%)	7 (12)	11 (28)	0.037
Average RASS score during the day shift from ICU day 1 to ICU day 3, median (IQR) ^b^	0 (0–0)	0 (−2–0)	0.070
Time to first out of bed mobilization after ICU admission (days)	1.7 (0.9–2.0)	5.3 (4.0–8.0)	<0.001
Highest ICU mobility scale score during ICU stay	8 (6–10)	4 (3–7)	<0.001
Number of daily rehabilitations per person on the ward (minute/time)	31 (22–43)	34 (27–40)	0.331

Data are presented as median (interquartile range) or number (%). IQR = interquartile range; BMI = Body mass index; ICU = Intensive Care Unit; APACHE = Acute Physiology and Chronic Health Evaluation; SOFA = Sequential Organ Failure Assessment; RASS = Richmond agitation sedation scale. ^a^ Barthel index before hospitalization was scored at the time of ICU admission based on the information from the family or the patients if they were conscious. ^b^ In all participating hospitals, RASS score, as a sedation scale, was monitored every 2 h during the day shift by nurses and recorded in the medical record. The best RASS score, which means the recorded number closest to zero during the day, of each day from days 1 to 3 was used to calculate the average value of RASS.

**Table 2 jcm-11-02587-t002:** Outcomes: psychiatric disorders in ICU survivors.

Outcomes	Early Mobilization Group (*n* = 60)	Non-Early Mobilization Group (*n* = 39)	*p* Value	Adjusted ^b^ *p* Value
Primary Outcome				
Follow-up at 3 months after discharge				
Patients with psychiatric symptoms, *n* (%) ^a^	15 (25)	20 (51)	0.008	0.032
At the time of hospital discharge				
Patients with psychiatric symptoms, *n* (%) ^a^	20 (33)	17 (46)	0.214	0.965
Secondary Outcomes				
Follow-up at 3 months after discharge				
Patients who scored HADS subset for depression ≥8, *n* (%)	12 (20)	15 (39)	0.044	0.107
HADS subset score for depression, median (IQR)	4 (2–7)	6 (2–9)	0.142	0.223
Patients who scored HADS subset for anxiety ≥8, *n* (%)	5 (8)	12 (30)	0.004	0.104
HADS subset score for anxiety, median (IQR)	3 (1–5)	6 (4–8)	<0.001	0.004
Patients who scored IES-R ≥25, *n* (%)	1 (2)	9 (23)	<0.001	0.026
IES-R score, median (IQR)	4 (1–9)	9 (4–20)	<0.001	0.009
At the time of hospital discharge				
Patients who scored HADS subset for depression ≥8, *n* (%)	16 (27)	15 (40)	0.155	0.917
HADS subset score for depression, median (IQR)	4 (2–8)	6 (4–9)	0.086	0.471
Patients who scored HADS subset for anxiety ≥8, *n* (%)	11 (18)	7 (19)	0.943	0.772
HADS subset score for anxiety, median (IQR)	3 (1–6)	4 (2–7)	0.392	0.448
Patients who scored IES-R ≥25, *n* (%)	2 (3)	5 (14)	0.102	0.266
IES-R score, median (IQR)	6 (2–12)	10 (4–17)	0.052	0.163
Changes between follow-up at 3 months and hospital discharge				
HADS subset score for depression, median (IQR)	1 (−2–3)	1 (−1–3)	0.928	0.418
HADS subset score for anxiety, median (IQR)	1 (−2–4)	−1 (−4–1)	0.006	0.032
IES-R score, median (IQR)	1 (−2–7)	−1 (−7–5)	0.053	0.131

Data are presented as number (%) or median (interquartile range). ICU = Intensive Care Unit, HADS = Hospital anxiety and depression scale, IQR = interquartile range, IES-R = Impact of event scale-revised, EQ-5D-5L = EuroQol-5 Dimensions-5 Levels. ^a^ Psychiatric symptoms were defined as the presence of at least one of three symptoms; depression, anxiety, and PTSD. ^b^ Multiple linear for continuous variable or multiple logistic regression analysis for nominal variables were performed to identify an association of the primary outcome with the following covariates. The covariates in the multi-variates analysis included age, male gender, Barthel index before hospitalization, ICU admission diagnosis (acute respiratory failure, cardiovascular disease, gastric or colonic surgery, sepsis, other), acute physiology and chronic health evaluation II score, use of mechanical ventilation, use of continuous analgesia, use of continuous sedation, use of steroids, use of neuromuscular blocking agents, and use of dialysis.

**Table 3 jcm-11-02587-t003:** Association between early mobilization and the presence of psychiatric symptoms.

Variables	Risk Ratio (95% CI)	Unadjusted Odds Ratio (95% CI)	Adjusted ^b^ Odds Ratio (95% CI)
Follow-up at 3 months after discharge			
Presence of psychiatric Symptoms ^a^	0.49 (0.29–0.83) *	0.32 (0.13–0.74) **	0.27 (0.08–0.89) *
HADS depression score ≥8	0.52 (0.27–0.99) **	0.40 (0.16–0.98) *	0.37 (0.11–1.24)
HADS anxiety score ≥8	0.27 (0.10–0.71) ***	0.20 (0.06–0.61) **	0.23 (0.06–1.31)
IES-R score ≥25	0.07 (0.01–0.54) ***	0.06 (0.01–0.32) ***	0.06 (0.01–0.70) *
At hospital discharge			
Presence of psychiatric Symptoms ^a^	0.73 (0.44–1.20)	0.59 (0.25–1.36)	1.03 (0.33–3.14)
HADS depression score ≥8	0.66 (0.37–1.16)	0.53 (0.22–1.27)	1.07 (0.29–3.91)
HADS anxiety score ≥8	0.97 (0.41–2.27)	0.96 (0.34–2.87)	1.23 (0.30–5.05)
IES-R score ≥25	0.25 (0.05–1.20)	0.22 (0.03–1.09)	0.01 (0.01–52.4)

The data are presented as risk ratio or odds ratio with 95% confidence interval. * <0.05, ** <0.01, *** <0.001. HADS = Hospital anxiety and depression scale, IES-R = Impact of event scale-revised, CI = Confidence interval. ^a^ Psychiatric symptoms are defined as the presence of at least one of three symptoms; depression, anxiety, and PTSD. ^b^ Multiple logistic regression analysis was performed to identify an association of the primary outcome with the following covariates. The covariates in the multi-variates analysis included age, male gender, Barthel index before hospitalization, ICU admission diagnosis (acute respiratory failure, cardiovascular disease, gastric or colonic surgery, sepsis, other), acute physiology and chronic health evaluation II score, use of mechanical ventilation, use of continuous analgesia, use of continuous sedation, use of steroids, use of neuromuscular blocking agents, and use of dialysis.

**Table 4 jcm-11-02587-t004:** Comparison of clinical outcomes between early mobilization group and non-early mobilization group.

Outcomes	Early Mobilization Group (*n* = 60)	Non-Early Mobilization Group (*n* = 39)	*p* Value	Adjusted ^a^ *p* Value ^a^
Health-related quality of life				
The EQ-5D-5L index at the time of 3 month after hospital discharge	0.89 (0.70–0.94)	0.82 (0.70–0.94)	0.235	0.952
The EQ-5D-5L index at the time of hospital discharge	0.81 (0.71–0.89)	0.70 (0.44–0.94)	0.384	0.926
Clinical outcomes and physical assessment				
The number of patients who can walk independently at the time of hospital discharge	58 (97)	33 (85)	0.032	0.136
Duration of mechanical ventilation (days)	1.4 (1–2)	4.4 (2–8.7)	<0.001	0.326
ICU length of stay (days)	4 (3–5)	7 (5–13)	<0.001	<0.001
Hospital length of stay (days)	22 (19–29)	32 (22–51)	0.004	0.009
Barthel index at hospital discharge	100 (90–100)	90 (70–100)	0.022	0.020
The number of patients who diagnosed as delirium during ICU stay	11 (18)	16 (41)	0.013	0.001
The number of patients who diagnosed as ICU acquired weakness at the time of ICU discharge	2 (3)	8 (21)	0.006	0.004

Data are presented as median (interquartile range) or number (%). IQR = interquartile range; ICU = Intensive Care Unit; IMS = ICU mobility scale. ^a^ Multiple linear for continuous variable or multiple logistic regression analysis for nominal variable were performed to identify an association of the primary outcome with the following covariates. The covariates in the multi-variates analysis included age, male gender, Barthel index before hospitalization, ICU admission diagnosis (acute respiratory failure, cardiovascular disease, gastric or colonic surgery, sepsis, other), acute physiology and chronic health evaluation II score, use of mechanical ventilation, use of continuous analgesia, use of continuous sedation, use of steroids, use of neuromuscular blocking agents and use of dialysis.

## Data Availability

The dataset generated and analyzed during the current study is not publicly available but is available from the corresponding author on reasonable request.

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
