# Peer review of "Association between Early Mobilization in the ICU and Psychiatric Symptoms after Surviving a Critical Illness: A Multi-Center Prospective Cohort Study"

_jcm, 2022, doi:10.3390/jcm11092587_

Round 1

Reviewer 1 Report

In this 9-center prospective observational study, the authors investigated the association between early mobilization and psychiatric symptoms. They concluded EM in ICU is significantly associated with a reduction in psychiatric symptoms at 3-month s after hospital discharge.

General comments:

Recent studies highlighted the importance of high mobility in early ICU stay. The authors evaluated an attractive topic, and the primary outcome is also a subject of high research value. I have no doubt that EM in ICU is associated with a low incidence of psychiatric symptoms at 3-month from this study data. My question is, can we say early mobilization "reduced" (as described in conclusion) or had an impact on (as described in the title) psychiatric symptoms from this data?

The major concern is the comparability of the two groups. As described in many feasibility studies of ICU mobilization, respiratory and hemodynamic instability could be a barrier to rehabilitation. Patients who sit at the edge of a bed in 3 ICU-day are less critical patients and probably have better functional status. High severity (Table 1) (with a higher number of painful procedures) and low functional status at discharge (Table 4) could be another cause of PTSD, anxiety, or depression after hospital discharge. High severity may also cause functional impairment (including muscle weakness) and reduce the quality of life after discharge. The EM feasibility could be just a prognostic indicator of better psychologic outcomes. I think authors need to support their argument in this respect.

1) Use of mechanical ventilation, continuous analgesia & sedation are not sufficient for adjustment of severity. I suggest adding other confounding factors indicating severity or functional status, such as APACH II score ICU admission diagnosis (ARDS, sepsis) in the multiple logistic regression. I also wonder about a low functional status at discharge mediated psychologic symptoms after discharge.

Abstract

2) Conclusion in the Abstract is different from the conclusion in the main text. "was associated" vs. "reduced." There is a huge difference. Please be consistent.

3) Authors mentioned IPTW methods. However, the result does not include specific values and numbers. Please describe it (such as risk ratio, confidence interval)

Introduction

4) (Page 2, Line 72) Authors described the study's purpose as "to investigate the relationship between rehabilitation and psychiatric symptoms.: Considering the previous comment 2), please be consistent with the conclusion (association vs. impact/reduction).

Methods

5) (Page 2, Line 92) I cannot find the supplementary table (Table S1)

6) (Page 2, Line 96) The definition of the non-EM group is confusing. It is described as the patients who did not receive EM. However, I think the non-EM group indicates patients who received rehabilitation after 72 hours. It is still not clear non-EM group is a mixture of delayed rehabilitation (72 hours) and never-rehabilitation group. Please clarify the definition.

7) (Page, Line 95-96) (Page 2, Line 97)(Page 4, Line 166-167) Mobilization is defined redundantly

8) (Page 4, Line 151 – 169) Description of secondary outcome is confusing.

  1. i) Incidence of depression, anxiety, and PTSD are described in both primary and secondary outcomes.
  2. ii) Outcomes usually indicate the affected parameters by the intervention/modification (in this study, early rehabilitation in 72 hours). The authors described intervention parameters (time to first out of bed mobilization, highest ICU mobility scale, number of daily rehabilitation sessions) as outcomes in the other outcome section. (Line 159 - 161). Consider describing those 3 parameters of Table 4 in a separate table or replace them in table 1.

iii) EQ-5D-5L …. Mechanical ventilation duration, length of ICU/hospital stay …. ICU acquired weakness, etc. may be affected by EM. However, they could be confounders of psychiatric symptoms. I think it is better to describe them as other variables/parameters or confounders measured.

Results

9) (Figure 1) Why did authors treat ICU deaths differently from other deaths after ICU discharge in figure 1 and IPTW analysis?

10) (Table 2, Table 3) Please describe statistical methods used in the adjusted model and variables adjusted in the analysis at the table using legend superscripts.

11) (Table 2, Table 3) Are the adjusted p-value of Table 2 and Table 3 the same? Adjusted p-values in both tables look redundant. On the other hand, some P-values have different values (0.035 vs. 0.032 of depression ≥8 follow up at 3 months; 0.267 vs. 0.266 of depression ≥8 at hospital discharge).

12) Please use an explanatory title in Table 4. Please describe the meaning of superscripts (a, b, c, d, e) or remove them.

13) (Page 8, Line 267-272) Please add specific values or numbers in the main text separately from the table.

Discussion

14) Consider multivariable analysis including physical impairment parameters to evaluate mediating effects of them. It will support or enrich the discussion arguments (lines 299-302 & line 322-334) (Page 9)

Minor Comment

15) (Page 4, Line 152) HADS is described as "HADs" only in this section. Please be consistent.

Thank you for the opportunity to review this article.

Author Response

Reviewer Comments and Response to Reviewers:

Reviewer #1

In this 9-center prospective observational study, the authors investigated the association between early mobilization and psychiatric symptoms. They concluded EM in ICU is significantly associated with a reduction in psychiatric symptoms at 3-month s after hospital discharge.

General comments:

Recent studies highlighted the importance of high mobility in early ICU stay. The authors evaluated an attractive topic, and the primary outcome is also a subject of high research value. I have no doubt that EM in ICU is associated with a low incidence of psychiatric symptoms at 3-month from this study data. My question is, can we say early mobilization "reduced" (as described in conclusion) or had an impact on (as described in the title) psychiatric symptoms from this data?

RESPONSE: We completely agree with your comments. Based on the design of this study, it is not appropriate to use the words such as "reduced" or "impact" that could imply the causal inference or relationship between Early Mobilization (EM) and the psychiatric symptoms at 3-months follow-up. Therefore, we revised the relevant parts thorough the manuscript including Title of this study. (Title, Page 1 / Lines 2-4) (Introduction, Page 2/Lines 72) (Conclusion, Page 12/Lines 368)

The major concern is the comparability of the two groups. As described in many feasibility studies of ICU mobilization, respiratory and hemodynamic instability could be a barrier to rehabilitation. Patients who sit at the edge of a bed in 3 ICU-day are less critical patients and probably have better functional status. High severity (Table 1) (with a higher number of painful procedures) and low functional status at discharge (Table 4) could be another cause of PTSD, anxiety, or depression after hospital discharge. High severity may also cause functional impairment (including muscle weakness) and reduce the quality of life after discharge. The EM feasibility could be just a prognostic indicator of better psychologic outcomes. I think authors need to support their argument in this respect.

RESPONSE: As you pointed out, the comparability of the two groups is the major limitation of this study. The severity and low functional status at discharge could be also confounding factors with the psychiatric symptoms after a critical illness that should be addressed in this study.

To adjust the severity of the two groups, we have re-analyzed all outcomes with additional covariates, such as APACHE II score, Barthel index before hospitalization, ICU admission diagnosis, and use of dialysis, neuromuscular blocking agents, and steroids. As you suggested, these need to be included in the analysis based on the results of the differences in baseline characteristics and previous papers. We added the references to support why these parameters should be included into Table S4. Even after we adjusted for these important and potential confounding factors including the severity, the primary outcome that EM is associated with a lower rate of psychiatric symptoms at 3-months follow-up were consistent in this study. We revised, Abstract (Page 2 / Line 49), the statistical method in Materials and Methods (Page 5 / Lines 182-186) and the p-values of Results and Tables 2 and 4, and Table S4, accordingly. Furthermore, we highlighted that we performed the statistical analysis in consideration of the differences in severity in the manuscript. (Results, Page 7 / Lines 249-250) (Page 11 / Lines 303-304)

Since the functional status at hospital discharge might be an intervening factor of the outcomes, we did not include the functional status at hospital discharge in the main analysis. If we include the functional status at hospital discharge, it might cause under/over-estimation of the association between EM and psychiatric symptoms after hospital discharge. Instead, we included the Barthel index before hospitalization in the analysis as a basic function of the patients. As an additional analysis we performed logistic regression analysis to investigate the association between functional status at hospital discharge and the psychiatric symptoms at the 3-months follow up as follows. This analysis shows that the functional status at hospital discharge is not associated with psychiatric symptoms at 3-months follow up.

Outcomes

Adjusted Odds ratio

(95ï¼…CI)

P value

Number of patients who could walk independently at the time of hospital discharge, n (%)

1.61 (0.17–15.11)

0.677

Barthel index at hospital discharge

0.98 (0.92–1.05)

0.634

Number of patients who diagnosed as ICU acquired weakness at the time of ICU discharge, n (%)

1.11 (0.20–6.21)

0.907

  Multiple logistic regression analysis for nominal variable were performed to determine the primary outcome with the following covariates. The covariates in the multi-variates analysis were selected age, male, barthel index before hospitalization, ICU admission diagnosis (acute respiratory failure, cardiovascular disease, gastric or colonic surgery, sepsis, other), acute physiology and chronic health evaluation II score, use of mechanical ventilation, use of continuous analgesia, use of continuous sedation, use of steroid, use of neuromuscular blocking agent, and use of dialysis.

We did not include this table because of the reasons described above. If we did not correctly understand the points raised by the reviewer, we would appreciate another opportunity to respond to them.

  • Use of mechanical ventilation, continuous analgesia & sedation are not sufficient for adjustment of severity. I suggest adding other confounding factors indicating severity or functional status, such as APACH II score ICU admission diagnosis (ARDS, sepsis) in the multiple logistic regression. I also wonder about a low functional status at discharge mediated psychologic symptoms after discharge.

RESPONSE: As you pointed out, the difference in the severity should be considered in multiple analysis. Therefore, we re-analyzed all outcomes with additional covariates indicating severity, such as APACHE II score, Barthel index before hospitalization, ICU admission diagnosis, and use of dialysis, neuromuscular blocking agents, and steroids. Even after we adjusted for these important and potential confounding factors including severity, the primary outcomes that EM is associated with a lower rate of psychiatric symptoms at 3-months follow-up are consistent in this study. We revised, Abstract (Page 2 / Line 49), the statistical method in Materials and Methods (Page 5 / Lines 182-186) and the p-values of Results and Tables 2 and 4, and Table S4, accordingly. Furthermore, we highlighted that we performed the statistical analysis in consideration of the differences in the severity in the manuscript. (Results, Page 7 / Lines 249-250) (Page 11 / Lines 303-304)

We understand your question regarding a low functional status at hospital discharge on psychiatric symptoms after a critical illness. As we discussed in the comment before, we did not include the functional status at hospital discharge in the main analysis because functional status at hospital discharge could be an intervening factor between EM and the psychiatric symptoms after a critical illness and cause under-/over-estimation of the association of interest. Furthermore, the additional analysis not included in the manuscript showed that functional status at hospital discharge was not associated with psychiatric symptoms at the 3-months follow up.

Abstract

2) Conclusion in the Abstract is different from the conclusion in the main text. "was associated" vs. "reduced." There is a huge difference. Please be consistent.

RESPONSE: We would like to thank the accurate and important comment raised by the reviewer. We completely agree with this comment and revised the relevant parts entirely through the manuscript include Conclusion according to the comment. (Title, Page 1 / Lines 2-4) (Introduction, Page 2/Lines 72) (Conclusion, Page 12/Lines 368)

3) Authors mentioned IPTW methods. However, the result does not include specific values and numbers. Please describe it (such as risk ratio, confidence interval)

RESPONSE: We added specific values and numbers, such as odds ratios and confidence intervals, according to your comments. Based on these comments, this abstract become more informative and understandable for readers. (ABSTRACT session, Page 2 / Lines 50-53)

Introduction

4) (Page 2, Line 72) Authors described the study's purpose as "to investigate the relationship between rehabilitation and psychiatric symptoms.: Considering the previous comment 2), please be consistent with the conclusion (association vs. impact/reduction).

RESPONSE: We revised the word “relationship” to “association” according to the comments (Introduction, Page 2 / Line 72)

Methods

5) (Page 2, Line 92) I cannot find the supplementary table (Table S1)

RESPONSE: The supplementary table (Table S1) was submitted as a separate file. We also added the Table S1 as below.

Table S1. Exclusion criteria

Exclusion Criteria

Details

Age <18 years

Unable to walk independently before ICU admission

Patients who require a wheelchair other than a cane or other walking assistance prior to admission or help from others to walk were considered unable to walk independently.

Neurological complications

Neurological complications include cerebral infarction, cerebral hemorrhage, acute subdual hematoma, acute epidural hematoma, traumatic subarachnoid hemorrhage, and encephalitis.

Lack of communication skill because of pre-existing mental diseases

Mental diseases include depression, anxiety, schizophrenia, dementia, cerebral infarction, cerebral hemorrhage, and alcoholism.

In terminal state or end of life

6) (Page 2, Line 96) The definition of the non-EM group is confusing. It is described as the patients who did not receive EM. However, I think the non-EM group indicates patients who received rehabilitation after 72 hours. It is still not clear non-EM group is a mixture of delayed rehabilitation (72 hours) and never-rehabilitation group. Please clarify the definition.

RESPONSE: We revised the sentences to define each group based on your comments. (Materials and Methods, Page 3 / Lines 91-96)

7) (Page, Line 95-96) (Page 2, Line 97)(Page 4, Line 166-167) Mobilization is defined redundantly

RESPONSE: We completely agree with this comment and revised the manuscript accordingly. The definitions of mobilization and each group are specifically described in the section “2.1 Study Design and Patient Selection” (Materials and Methods, Page 3 / Lines 91-100)

8) (Page 4, Line 151 – 169) Description of secondary outcome is confusing.

  1. Incidence of depression, anxiety, and PTSD are described in both primary and secondary outcomes.

RESPONSE: We greatly appreciate your precise review. We removed the sentence from the secondary outcomes accordingly.

  1. ii) Outcomes usually indicate the affected parameters by the intervention/modification (in this study, early rehabilitation in 72 hours). The authors described intervention parameters (time to first out of bed mobilization, highest ICU mobility scale, number of daily rehabilitation sessions) as outcomes in the other outcome section. (Line 159 - 161). Consider describing those 3 parameters of Table 4 in a separate table or replace them in table 1.

RESPONSE Based on your comments, we have replaced three parameters: time to first out of bed mobilization, the highest ICU mobility scale, and number of daily rehabilitation sessions, as parameters in table 1. We also revised the relevant sentences in the manuscript accordingly. (Materials and Methods, Page 4 / Lines 136-138, 141-143), (Results, Page 5/ Lines 212-214), and Table 1

iii) EQ-5D-5L …. Mechanical ventilation duration, length of ICU/hospital stay …. ICU acquired weakness, etc. may be affected by EM. However, they could be confounders of psychiatric symptoms. I think it is better to describe them as other variables/parameters or confounders measured.

 RESPONSE: Based on your comments, we revised "other outcome" to “other variables/parameters”. As we discussed before, these parameters could be confounding factors for the primary outcomes. However, adjusting for these parameters, which are intervening factors, might result in under-/over-estimation of the findings. Therefore, we did not include the further analysis with adjustment for these parameters in the main analysis. (Materials and Methods, Page 4 / Lines 157) and (Results, Page 9 / Line 281)

Results

9) (Figure 1) Why did authors treat ICU deaths differently from other deaths after ICU discharge in figure 1 and IPTW analysis?

RESPONSE: Given the definition of the post-intensive care syndrome (PICS), we targeted patients who survived the ICU in this study. Therefore, death during the ICU stay was excluded from the analysis. However, excluding death during the ICU stay could cause selection bias because death during the ICU stay could be a competing risk factor. We added text to explain these in the limitations (Discussions, Page 12 / Lines 362-365)

10) (Table 2, Table 3) Please describe statistical methods used in the adjusted model and variables adjusted in the analysis at the table using legend superscripts.

RESPONSE: We added a description of the statistical methods used in the adjusted model and the variables in the analysis at the legends of Table 2, 3, and 4, and Table S7, accordingly.

11) (Table 2, Table 3) Are the adjusted p-value of Table 2 and Table 3 the same? Adjusted p-values in both tables look redundant. On the other hand, some P-values have different values (0.035 vs. 0.032 of depression ≥8 follow up at 3 months; 0.267 vs. 0.266 of depression ≥8 at hospital discharge).

RESPONSE: We agree with your comment that duplicating p-values is redundant. We removed the p-values from Table 3 to make the message from the Table concise.

12) Please use an explanatory title in Table 4. Please describe the meaning of superscripts (a, b, c, d, e) or remove them.

RESPONSE: We greatly appreciate your precise review. We removed the superscripts (a, b, c, d, e) in Table 4, accordingly

13) (Page 8, Line 267-272) Please add specific values or numbers in the main text separately from the table.

 RESPONSE: We added specific values and numbers to the main manuscript separately from the tables accordingly. (Results, Page 7 / Lines 251-260, Page 8-9 / Line 264-268, 276-278, Page 7 / Lines 283-287, and 293-294)

Discussion

14) Consider multivariable analysis including physical impairment parameters to evaluate mediating effects of them. It will support or enrich the discussion arguments (lines 299-302 & line 322-334) (Page 9)

RESPONSE: We greatly appreciate your comments. As we discussed in response to your previous comments, we are concerned about under-/over-estimation of the association between EM and the primary outcome by including intervening factors in the analysis, such as physical impairment parameters at the time of hospital discharge. To make the message from our findings clear and simple, we did not include the analysis with adjustment for the physical status at hospital discharge. Instead, we included the basic physical function (Barthel Index before hospitalization) as a covariate before starting the exposure of mobilization or rehabilitation. Furthermore, as you kindly advised, we added these potential mediating effects to the Discussion section accordingly (Page 12 / Lines 343-348.).

Minor Comment

15) (Page 4, Line 152) HADS is described as "HADs" only in this section. Please be consistent.

RESPONSE: We revised the word accordingly. (Results session, Page 4 / Line 154)

Reviewer 2 Report

Thanks you for the opportunity to review this interesting manuscript written by dr Shinichi Watanabe et al. titled "Impact of early mobilization in the ICU on psychiatric symptoms after surviving a critical illness: A multi-center prospective cohort study"

The manuscript focuses on to investigate the association between early mobilization (EM), defined as a rehabilitation level of sitting at the edge of the bed or higher within 72-hours of ICU admission, and psychiatric outcomes at 3-months after hospital discharge of consecutive patients, between June and December 2019, admitted to the ICU for more than 48 hours.

The manuscript is well written and I congratulate the authors for a prospective multicenter study. However, there are some major points that I would like the authors to discuss further.

Why the study did not provide randomization of the enrollment? please add the explanation in the methods and explain it in more details as it was mentioned only in the limitations of the study but it is very important bias of the study.

In Table 1 among the baseline characteristics of the patients there are 

The use of steroid during ICU stay, n (%) 7 (12) 14 (35) 0.006
The use of neuromuscular blocking agent during ICU stay, n (%) 0 (0) 5 (13) 0.008
The use of dialysis during ICU stay, n (%) 7 (12) 11 (28) 0.037

I disagree with the authors the there are no differences among the two groups as these findings reach statistical significance. please comment them in the results and discussion.

Moreover, these values are higher in the non EM group vs the EM group. please comment these findings as they could alone predispose to the failure of the primary outcomes found in the non EM group. Indeed the longer use of steroids, higher use of neuroblocking agents and dialysis may have predisposed to worsening admission clinical outcomes and  incidence of depression, anxiety and the HADS subset scores for depression and anxiety. Did you perform a specific analysis (multiple regression analysis) to understand the correlations among all these factors?  please comment.

Thanks again for the opportunity to revise this manuscript.

I look forward to read the authors' feedback

Author Response

Reviewer #2

Thanks you for the opportunity to review this interesting manuscript written by dr Shinichi Watanabe et al. titled "Impact of early mobilization in the ICU on psychiatric symptoms after surviving a critical illness: A multi-center prospective cohort study"

The manuscript focuses on to investigate the association between early mobilization (EM), defined as a rehabilitation level of sitting at the edge of the bed or higher within 72-hours of ICU admission, and psychiatric outcomes at 3-months after hospital discharge of consecutive patients, between June and December 2019, admitted to the ICU for more than 48 hours.

The manuscript is well written, and I congratulate the authors for a prospective multicenter study. However, there are some major points that I would like the authors to discuss further.

RESPONSE: We thank the reviewer for the comprehensive review and positive comments.

Why the study did not provide randomization of the enrollment? please add the explanation in the methods and explain it in more details as it was mentioned only in the limitations of the study but it is very important bias of the study.

RESPONSE: We agree with you that randomization is very important to conclude about the effects of EM on outcomes. As this study design was a prospective observational study, we could not randomize the patients into two groups. Therefore, we did two things. First, we revised the text that could imply a causal inference or relationship between the exposure, which is early mobilization (EM) in this study, and the outcomes through the manuscript to prevent readers from misunderstanding the findings. Furthermore, we revised title of this study from “Impact” that could imply the causal inference to “association”. Second, we added a sentence to explain why this study did not randomize the patients and how the exposure of EM was allocated in this study. (Materials and Methods, Page 3 / Lines 96-100)

In Table 1 among the baseline characteristics of the patients there are

The use of steroid during ICU stay, n (%) 7 (12) 14 (35) 0.006

The use of neuromuscular blocking agent during ICU stay, n (%) 0 (0) 5 (13) 0.008

The use of dialysis during ICU stay, n (%) 7 (12) 11 (28) 0.037

I disagree with the authors the there are no differences among the two groups as these findings reach statistical significance. please comment them in the results and discussion.

RESPONSE: We greatly appreciate your precise review. As you pointed out, there are significant differences in the baseline characteristics of the two groups. We revised the relevant sentences correctly based on the results in the Tables. (Results, Page 5 / Lines 207-214) (Discussions, Page 12 / 354-356)

Moreover, these values are higher in the non EM group vs the EM group. please comment these findings as they could alone predispose to the failure of the primary outcomes found in the non EM group. Indeed the longer use of steroids, higher use of neuroblocking agents and dialysis may have predisposed to worsening admission clinical outcomes and  incidence of depression, anxiety and the HADS subset scores for depression and anxiety. Did you perform a specific analysis (multiple regression analysis) to understand the correlations among all these factors?  please comment.

RESPONSE: Based on your comments and other reviewers’ comments, we found that these significant differences in important baseline characteristics should be adjusted in the primary analysis to prevent the over-estimation of the findings. Therefore, we re-analyzed all multivariate analysis by including the following important characteristics as covariates: APACHE II score, Barthel index before hospitalization, ICU admission diagnosis, and use of dialysis, neuromuscular blocking agent, and steroid. We also added the newly calculated adjusted p-values to Tables 2 and 4 accordingly. Of note, the results were consistent even after adjustment of the significant differences raised by the reviewer. To describe clearly what covariates were used in the multivariate analysis, we added an explanation of the statistical methos in Material and Methods section (Page 5 / Lines 182-186) and the legends in Table 2-4 and Table S7.

Reviewer 3 Report

This is an interesting study about the potential benefits of early mobilization (EM) in the prevention of possible mental health issues for patients admitted to ICUs. Authors found that patients who were administered EM had lower risk of experiencing psychiatric symptoms (specifically, 0.49 times) than those who did not receive it. However, the study has important drawbacks discussed below:

  1. Introduction

I recognize that minimizing space and avoiding excessive wording are noble intents. However, there are very few references that support the theoretical background. For example, the first one (Bolton et al., 2007), did not actually say that there is a relationship between critical illness and psychiatric disorders. In fact, no mention of any psychiatric disorder was made in this article. Besides, the one on which this study is primarily based (Kume et al., 2004) is not available in English (not even the abstract) and that is a problem, because it constrains the capacity of non-Japanese speakers to undertake a critical review of the literature.

  1. Materials and methods

 2.1 Study design and patient selection

The randomization is obviously a very important element in this type of studies.

Additionally, I am particularly concerned about the difference between groups in the use of steroid, neuromuscular blocking agent and dialysis at baseline. Significantly more patients in the non-EM group (in some cases, even double) needed these treatments compared to the EM group. I do not know what implications these variables have on psychiatric symptoms; however, I would say that patients who need the use of these treatments present more serious illnesses than those who not. I wonder if this could have an influence on the results obtained.

In my opinion, based on these shortcomings groups may not be comparable, and results are highly debatable.

 2.2 Statistical analysis

I wonder why a non-parametric test such as the Mann-Whitney test is used with a sample of 240 subjects.

I see in the results section that adjusted p-values are provided. However, no mention to the adjustment method is made in the manuscript.

  1. Results

This section is sometimes difficult to follow considering the number of tables. However, essential information is lacking. I would include the value of the statistics such as the beta coefficients resulting from the regression analyses. This would help the interpretation of the outcomes.

Some information (for instance, risk ratios) is duplicated in both the text and tables. Maybe eliminating any duplication would help lighten the results section.

Minor concerns

Supplementary material

Table S9: remove the s from “psychiatrics” in the two first rows. Aldo, in the footnote: Psychiatric symptoms was were.

Author Response

Reviewer #3

This is an interesting study about the potential benefits of early mobilization (EM) in the prevention of possible mental health issues for patients admitted to ICUs. Authors found that patients who were administered EM had lower risk of experiencing psychiatric symptoms (specifically, 0.49 times) than those who did not receive it. However, the study has important drawbacks discussed below:

  1. Introduction I recognize that minimizing space and avoiding excessive wording are noble intents. However, there are very few references that support the theoretical background. For example, the first one (Bolton et al., 2007), did not actually say that there is a relationship between critical illness and psychiatric disorders. In fact, no mention of any psychiatric disorder was made in this article. Besides, the one on which this study is primarily based (Kume et al., 2004) is not available in English (not even the abstract) and that is a problem, because it constrains the capacity of non-Japanese speakers to undertake a critical review of the literature.

RESPONSE: We greatly appreciate your suggestion. We carefully reviewed the Introduction and put relevant references in the appropriate part of Introduction. We removed (Bolton et al., 2007) and (Kume et al., 2004) according to your comments. Furthermore, we added the following references to the Introduction to support our theoretical background.

Additional references

1)Kawakami, D.; Fujitani, S.; Morimoto, T., Dote, H.; Takita, M.; Takaba, A.; Hino, M.; Nakamura, M.; Irie, H.; Adachi, T.; et al. Prevalence of post-intensive care syndrome among Japanese intensive care unit patients: a prospective, multicenter, observational J-PICS study. Critical care (London, England). 2021, 25(1), 69. DOI: 10.1186/s13054-021-03501-z

2)Jackson, J. C.; Pandharipande, P. P.; Girard, T. D.; Brummel, N. E.; Thompson, J. L.; Hughes, C. G.; Pun, B. T.; Vasilevskis, E. E.; Morandi, A.; Shintani, A. K.; et al. Depression, post-traumatic stress disorder, and functional disability in survivors of critical illness in the BRAIN-ICU study: a longitudinal cohort study. The Lancet. Respiratory medicine. 2014, 2(5), 369–379. https://doi.org/10.1016/S2213-2600(14)70051-7

8)Styf, J. R.; Hutchinson, K.; Carlsson, S. G.; Hargens, A. R. Depression, mood state, and back pain during microgravity simulated by bed rest. Psychosomatic medicine. 2001, 63(6), 862–864. https://doi.org/10.1097/00006842-200111000-00002

9)Ishizaki, Y.; Ishizaki, T.; Fukuoka, H.; Kim, C. S.; Fujita, M.; Maegawa, Y.; Fujioka, H.; Katsura, T.; Suzuki, Y.; Gunji, A. Changes in mood status and neurotic levels during a 20-day bed rest. Acta astronautica. 2002, 50(7), 453–459. https://doi.org/10.1016/s0094-5765(01)00189-8

  1. Materials and methods

2.1 Study design and patient selection The randomization is obviously a very important element in this type of studies. Additionally, I am particularly concerned about the difference between groups in the use of steroid, neuromuscular blocking agent and dialysis at baseline. Significantly more patients in the non-EM group (in some cases, even double) needed these treatments compared to the EM group. I do not know what implications these variables have on psychiatric symptoms; however, I would say that patients who need the use of these treatments present more serious illnesses than those who not. I wonder if this could have an influence on the results obtained. In my opinion, based on these shortcomings groups may not be comparable, and results are highly debatable.

RESPONSE: We completely agree with you that randomization is a very important element to conclude about the effect of EM on outcomes. Since this was a prospective observational study, we could not randomize the patients into two groups. Therefore, we did two things. First, we revised sentences that could imply a causal inference or relationship between the exposure, which is early mobilization (EM) in this study, and outcomes throughout the manuscript to prevent readers from misunderstanding these findings. Furthermore, we revised the title of this study from “Impact” that could imply the causal inference to “association”. Second, we added text to explain that this study did not randomize the patients and how the exposure of EM was allocated in this study. (Materials and Methods, Page 3 / Lines 96-100)

In addition to these revisions, we also totally agree with you that there are significant differences in the very important baseline characteristics as you raised that should be adjusted in the primary analysis to prevent the over-estimation of the findings. Therefore, we re-analyzed all multivariate analysis by including the following important characteristics as covariates: APACHE II score, Barthel index before hospitalization, ICU admission diagnosis, and use of dialysis, neuromuscular blocking agent, and steroid. We also added the newly calculated adjusted p-values to Tables 2 and 4 accordingly. Of note, the results were consistent even after adjustment of the significant differences raised by the reviewer. To describe clearly what covariates were used in the multivariate analysis, we added an explanation of the statistical methods in the Materials and Methods section (Page 5 / Lines 182-186) and the legends in Table 2-4 and Table S7. However, the results may still be controversial. We highlighted the necessity of the randomized controlled trial in the future in Discussion (Page 12 / Lines 362-365)

2.2 Statistical analysis I wonder why a non-parametric test such as the Mann-Whitney test is used with a sample of 240 subjects. I see in the results section that adjusted p-values are provided. However, no mention to the adjustment method is made in the manuscript.

RESPONSE: Before using non-parametric tests, we reviewed the distribution of each parameter with the Shapiro-Wilk test. There are only non-parametric variables, hence we used non-parametric tests in this analysis. We added text to explain how we confirmed the distribution of each parameter in the section “2.6 Statistical Analysis”. (Materials and Methods, Page 5 / Lines 175-176)

Furthermore, to describe clearly what covariates were used in the multivariate analysis, we added an explanation of the statistical methods to the Materials and Methods section (Page 5 / Lines 182-186) and the legends in Table 2-4 and Table S7 according to your comments.

  1. Results

This section is sometimes difficult to follow considering the number of tables. However, essential information is lacking. I would include the value of the statistics such as the beta coefficients resulting from the regression analyses. This would help the interpretation of the outcomes. Some information (for instance, risk ratios) is duplicated in both the text and tables. Maybe eliminating any duplication would help lighten the results section.

RESPONSE: In this study, logistic regression analysis is performed for nominal variables and multilinear analysis is performed for continuous variables. Therefore, we added the beta coefficients, which is important to interpret the outcomes as you mentioned, to Table S7 because the multilinear analysis was performed in Table S7. In addition, we thank you for the proposal to lighten the results section by eliminating any duplication. Since other two reviewers asked me to put the exact numbers and values obtained from the statistical analysis into the text, we had to add some key information showed in both the manuscript and the tables, we carefully reviewed and removed less important information of numbers and values through “Results”.

Minor concerns

Supplementary material Table S9: remove the s from “psychiatrics” in the two first rows. Aldo, in the footnote: Psychiatric symptoms was were

RESPONSE: We revised the relevant sentences accordingly.

Round 2

Reviewer 1 Report

General comments

Authors revised the manuscript well, according to the reviewer’s comments.

Concerns about methodology and conclusion have been greatly reduced.

However, statistical descriptions are causing confusion.

  1. (Page 1 Line 45): Multiple logistic regression … baselines and risk ratios (RR) was used.

It sounds like authors used multivariable regression for adjusted RR. In addition, “adjusted with baselines” have no specific information. Consider describe it as “Risk ratio (RR) and multiple logistic regression analysis were used.”

  1. (Line 49) “(25% vs 51%, adjusted p-0.032)”

This adjusted p-value is a statistical description of odds ratio, not a p-value of statistical comparison between 25% and 51%. Please consistent with statistical methods and p-value. (25% vs 51%, p-value 0.008). or (OR 0.27, p-value 0.032).

  1. Same confusion appears in results section (Line 248, Line 251, 253, 254, Line 283-287). Please check all adjusted p-values which did not described with odds ratio.
    4. (Line 274-279) “Multiple linear regression…. At 3-months follow-up.”

I think authors confused the statistical description of multiple linear regression and logistic regression. If authors used multiple linear regression, please describe BETA and p-value.

  1. (Line 91-93): Hard to read this sentence. Please modify this sentence.
  2. Please modify the title of Table 4 to better describe the content.

For example, comparison of clinical outcomes between EM group and non-EM group.

Minor

  1. (Line 157) Please select & use single term (Other variables vs. Other parameters)

Author Response

Reviewer #1

Authors revised the manuscript well, according to the reviewer’s comments.

Concerns about methodology and conclusion have been greatly reduced.

 However, statistical descriptions are causing confusion.

 (Page 1 Line 45): Multiple logistic regression … baselines and risk ratios (RR) was used.

It sounds like authors used multivariable regression for adjusted RR. In addition, “adjusted with baselines” have no specific information. Consider describe it as “Risk ratio (RR) and multiple logistic regression analysis were used.”

RESPONSE: We revised the text according to your comments.

(ABSTRACT, Page 1 / Lines 45-46)

(Line 49) “(25% vs 51%, adjusted p-0.032)”

This adjusted p-value is a statistical description of odds ratio, not a p-value of statistical comparison between 25% and 51%. Please consistent with statistical methods and p-value. (). or (OR 0.27, p-value 0.032).

Same confusion appears in results section (Line 248, Line 251, 253, 254, Line 283-287). Please check all adjusted p-values which did not described with odds ratio.

  1. (Line 274-279) “Multiple linear regression…. At 3-months follow-up.”

I think authors confused the statistical description of multiple linear regression and logistic regression. If authors used multiple linear regression, please describe BETA and p-value.

RESPONSE: We appreciate your suggestions. We revised the relevant sentences according to your comments. Regarding the adjusted p-values in Lines 283-287, we removed all adjusted-p vales since we did not describe it with an OR and these results were less important than the primary outcomes. (ABSTRACT, Page 2 / Lines 49-52), (Results section, Page 7 / Lines 249-253 and 275-277).

We revised the description of the results in “Multiple linear regression” with beta and p-values. (Results section, Page 9 / Lines 275-277)

(Line 91-93): Hard to read this sentence. Please modify this sentence.

Please modify the title of Table 4 to better describe the content.

For example, comparison of clinical outcomes between EM group and non-EM group.

 RESPONSE: We revised the relevant part of the manuscript (Materials and Methods section, Page 3 / Lines 91-93) and the title of Table 4 (Page 10 / Lines 284).

Minor

(Line 157) Please select & use single term (Other variables vs. Other parameters)

RESPONSE: We used “Other variables” according to your comment. (Materials and Methods section, Page 4 / Lines 158), (Results section, Page 9 / Lines 279)

Reviewer 2 Report

thanks for the opportunity to revise again this manuscript.

It is imporved. I would only suggest to underline better in the MM section the observational nature of the study.

I would also suggest a Mother tongue revision of the manuscript that should be acknowledged with initials in the acknowledgment session  .
thank you.

Author Response

Reviewer #2

thanks for the opportunity to revise again this manuscript.

It is improved. I would only suggest to underline better in the MM section the observational nature of the study.

I would also suggest a Mother tongue revision of the manuscript that should be acknowledged with initials in the acknowledgment session  .

thank you.

RESPONSE:

We greatly appreciate your review and comment. This manuscript was thoroughly reviewed and revised by one of the co-authors, Professor Alan Kawarai Lefor who is a native English speaker and well-published surgical scientist. While there may be an occasional typographic error or stylistic difference, we believe that this manuscript conforms to standard scientific English language usage. We would certainly be willing to address any specific issues that you identify.
